# Flagellar energy costs across the tree of life

**Paul E Schavemaker\*, Michael Lynch**

Biodesign Center for Mechanisms of Evolution, Arizona State University, Tempe, United States

**Abstract** Flagellar-driven motility grants unicellular organisms the ability to gather more food and avoid predators, but the energetic costs of construction and operation of flagella are considerable. Paths of flagellar evolution depend on the deviations between fitness gains and energy costs. Using structural data available for all three major flagellar types (bacterial, archaeal, and eukaryotic), flagellar construction costs were determined for *Escherichia coli*, *Pyrococcus furiosus*, and *Chlamydomonas reinhardtii*. Estimates of cell volumes, flagella numbers, and flagellum lengths from the literature yield flagellar costs for another ~200 species. The benefits of flagellar investment were analysed in terms of swimming speed, nutrient collection, and growth rate; showing, among other things, that the cost-effectiveness of bacterial and eukaryotic flagella follows a common trend. However, a comparison of whole-cell costs and flagellum costs across the Tree of Life reveals that only cells with larger cell volumes than the typical bacterium could evolve the more expensive eukaryotic flagellum. These findings provide insight into the unsolved evolutionary question of why the three domains of life each carry their own type of flagellum.

## Editor's evaluation

This work demonstrates convincingly that energetic considerations (building costs versus potential benefit) must be taken into account to understand the evolution of flagella. It provides compelling evidence to the long-standing question of why bacteria, archaea, and eukaryotes evolved with different types of flagella.

\*For correspondence:
pschavem@asu.edu

**Competing interest:** The authors declare that no competing interests exist.

## Introduction

Life contains a dazzling diversity of molecular and cellular mechanisms. These are the consequence of, and are subject to, evolution. Though speculation abounds, there is yet no full integration of molecular and cellular biology with evolutionary theory (*Lynch et al., 2014*; *Lynch and Trickovic, 2020*). All features of the cell require energy for construction and operation, but they differ in their energy demand and contribution to fitness. Energy costs act as weights to evolutionary paths. If we want to know why only certain molecular and cellular features exist, we need an accounting of their baseline costs. Such an accounting has been under way for individual genes (*Lynch and Marinov, 2015*), membranes (*Lynch and Marinov, 2017*), and various other features (*Lynch and Trickovic, 2020*; *Milo and Phillips, 2016*; *Raven and Richardson, 1984*). Many single-celled species swim by virtue of their flagella. However, why the three domains of life – bacteria, archaea, and eukaryotes – each evolved their own type of flagellum is still a major open evolutionary question.

The bacterial flagellum (*Berg, 2003*) consists of a helical protein filament, sometimes sheathed by a membrane (*Geis et al., 1993*), that is attached, via a hook region, to a basal body embedded in the cell envelope. The filament is assembled by progressive addition of monomers of the protein flagellin (FliC) to the tip, reached by diffusion through a central channel in the filament. Its rotation

**eLife digest** Most creatures on Earth are single cell organisms. The tree of life comprises three domains, two of which – bacteria and archaea – are formed exclusively of creatures that spend their existence as independent cells. Yet even eukaryotes, the domain which include animals and plants, feature single cell species such as yeasts and algae.

Regardless of which group they belong to, all single-celled organisms must find food in their environment. For this, many are equipped with flagella, whip-like structures that protrude from the cell and allow it to swim. In fact, archaea, bacteria and eukaryotes have all independently evolved these structures.

However, flagella are also expensive for an organism to build, maintain and operate. They are only worth having if the advantages they bring to the cell compensate for their cost; many single-cell species do not carry flagella and obtain their food without having to swim.

To explore this trade-off, Schavemaker and Lynch calculated the cost of building and using flagella for about 200 species across the tree of life. The analysis show that the amount of energy spent on flagella varied between 0.1% and 40% of the entire cell budget. This investment is only worthwhile if the cell is above a certain size. Smaller than this, and the organism is better off obtaining its food passively.

The results also show that while eukaryotic flagella are much bigger and quite different than their bacterial counterpart, both appendages share the same patterns of cost effectiveness. However only eukaryotic cells, which are on average larger than bacteria, can afford to evolve such sizable and complex structures; making just one would cost more than the entire energy budget of a bacterial cell.

Many single-cell species which are critical for the health of the planet are equipped with flagella, such as the microorganisms which recycle matter in the oceans and release carbon dioxide. Understanding the costs and benefits of flagella could explain more about this aspect of the carbon cycle, and therefore global warming.

---

is driven by a proton or sodium gradient (**Ito and Takahashi, 2017**). The archaeal flagellum, or archaellum, is also a rotating flagellum but differs from the bacterial one in two key respects. The filament, lacking a central channel, is assembled from the base, and its rotation is driven by ATP hydrolysis (**Albers and Jarrell, 2018**). The eukaryotic flagellum, or cilium, is completely different from the other two. Its diameter is an order of magnitude larger (**Khan and Scholey, 2018**); it bends rather than rotates; and the ATP-dependent bending is caused by motor proteins arranged along the length of the flagellum. Most eukaryotic flagella have two single microtubules in the middle, surrounded by nine doublet microtubules. The microtubules are connected by protein complexes such as inner and outer dynein arms (**King, 2016**), nexin-DRC (**Heuser et al., 2009**), and radial spokes (**Pigino et al., 2011**). This whole protein complex, the axoneme, is always surrounded by a membrane. Like the bacterial flagellum, the eukaryotic axoneme is assembled at the tip, but with subunit delivery being facilitated by active intraflagellar transport (IFT) (**Marshall and Rosenbaum, 2001**).

Flagellar construction and operating costs have been estimated for the bacterium *Escherichia coli* (**Macnab, 1996**) and a dinoflagellate eukaryote (**Raven and Richardson, 1984**). For *E. coli*, the construction and operating costs are 2% and 0.1%, respectively, relative to total cell energy expenditure (**Macnab, 1996**). Others found an operating cost of 3.4% (assuming 5 flagella, a 100 Hz rotation rate, and a 1 hr cell division time) (**Ziegler and Takors, 2020**). The relative construction cost for the dinoflagellate was estimated at 0.026%. The relative operating cost was 0.08–0.9%, compared to total cell maintenance cost (**Lynch and Marinov, 2015**; **Raven and Richardson, 1984**).

Here, following on previous work (**Lynch and Trickovic, 2020**), we provide a fuller accounting of construction costs for model bacterial, archaeal, and eukaryotic flagella, and extend these results to a range of bacterial and eukaryotic species using information on flagellum number, flagellum length, and cell volume retrieved from the literature. Drawing from additional estimates of the operating costs of flagella, these results are discussed in the context of swimming speed, nutrient uptake, growth rate, scaling laws, effective population size, and the origin of the eukaryotic flagellum.

# Results

## Energy costs of flagellum construction

Flagellar motility burdens the cell with costs of construction and operation. The former refers to the energy required for synthesis of the proteins and lipids that constitute the flagellum, which includes both direct and opportunity costs (**Lynch and Marinov, 2015**; **Mahmoudabadi et al., 2019**). The operating cost is the energy associated with rotating (bacteria and archaea) or beating the flagellum (eukaryotes).

Starting from the framework previously outlined (**Lynch and Marinov, 2015**), and ignoring protein turnover (which is only a minor contribution to costs; **Lynch and Marinov, 2015**; Methods), we estimate the construction cost per protein component (in units of ATP hydrolyses) as:

$$C_{prot} = C_{AA}N_pL_p, \tag{1}$$

where $N_p$ is the number of protein copies, $L_p$ is the length of the protein in amino acids, and $C_{AA}$ is the average energy cost of an amino acid (29 ATP; direct + opportunity costs) (**Lynch and Marinov, 2015**). When only the volume of a protein is known, we calculate the number of amino acids using the average volume per amino acid ($1.33 \times 10^{-10}$ µm³; Methods).

Eukaryotic flagella, and those of some bacteria, are enveloped by a membrane. Previously established approaches (**Lynch and Marinov, 2017**), with the inclusion of membrane proteins, leads to the membrane construction cost:

$$C_{mem} = \frac{2C_LA_{tot}\left(1-f_{prot}\right)}{A_{liphead}} + \frac{C_{AA}t_{mprot}A_{tot}f_{prot}}{V_{AA}}. \tag{2}$$

For the first term, $C_L$ is the average energy cost of a single lipid molecule, $A_{tot}$ is the total membrane surface area, which includes both lipids and membrane proteins, $A_{liphead}$ is the cross-sectional area of a lipid head-group, $f_{prot}$ is the fraction of membrane surface area occupied by protein (which is assumed

**Table 1.** Energy costs of flagella in the three domains of life.

Note that ATP denotes the number of ATP hydrolyses. A breakdown of the construction cost is available in *Table 1—source data 1*.

| | Bacteria | Archaea | Eukaryotes |
|---|---|---|---|
| Species | *Escherichia coli* | *Pyrococcus furiosus** | *Chlamydomonas reinhardtii* |
| Construction cost per µm (ATP) | $3.02 \times 10^7$ | $1.28 \times 10^7$ | $2.80 \times 10^9$ |
| Construction cost per flagellum (ATP) | $2.32 \times 10^8$ | $2.15 \times 10^7$ | $3.08 \times 10^{10}$ |
| Number of flagella per cell | 3.4 | 50 | 2 |
| Construction cost of all flagella (ATP cell⁻¹) | $7.88 \times 10^8$ | $1.07 \times 10^9$ | $6.15 \times 10^{10}$ |
| Cell volume (µm³) | 1.0 | 0.22 | 122 |
| Cell division time (hr) | 1.0 | 1.0 | 9.15 |
| Total cost of cell (construction + operating; ATP) | $1.59 \times 10^{10}$ | $6.50 \times 10^9$ | $5.32 \times 10^{12}$ |
| Relative construction cost, all flagella (%) | 5.0 | 16.5 | 1.4 |
| Operating cost per flagellum (ATP s⁻¹) | $6.6 \times 10^4$ | $2.64 \times 10^2$ [†] | $9.7 \times 10^5$ |
| Operating cost per cell cycle, all flagella (ATP) | $8.08 \times 10^8$ | $4.75 \times 10^7$ [††] | $6.39 \times 10^{10}$ |
| Relative operating cost, all flagella (%) | 5.2 | 0.73 [†] | 1.2 |
| Relative total cost, all flagella (%) | 10.2 | 17.2 [†] | 2.6 |

*Due to gaps in knowledge of *P. furiosus* flagella, some data were taken from other archaea (see main text).

[†]This estimate for the operating cost is probably too low as the flagellar rotation rate that it is based on was recorded with a bead attached, which slows down flagellar rotation.

The online version of this article includes the following source data for table 1:

**Source data 1.** Breakdown of flagellar construction costs for bacteria, archaea, and eukaryotes.

to be 0.4) (*Lindén et al., 2012*), and the factor 2 accounts for the bilayer of lipids. For the second term, $C_{AA}$ is the energy cost per amino acid, $t_{mprot}$ is the thickness of the protein areas of the membrane (8 nm), and $V_{AA}$ is the average volume per amino acid.

*Equations 1 and 2* yield estimates of *absolute* construction costs, in number of ATPs. *Relative* construction costs are obtained by dividing the absolute flagellum costs by the construction cost of the entire cell obtained from *Lynch and Marinov, 2015*. For a discussion of assumptions and omitted costs, see Methods. The flagellar construction cost data for the three model organisms are summarised in *Table 1*.

## Bacteria – cost of the *E. coli* flagellum

For a detailed examination of the energy costs of the bacterial flagellum, we chose that of *E. coli*. The protein composition of the flagellum, determined by a combination of structural and biochemical work, is summarised in *Berg, 2003*. We supplemented this with copy numbers for export-apparatus proteins (*Fukumura et al., 2017*; *Minamino, 2014*; *Minamino, 2018*) and verified the FliC copy number (*Namba et al., 1989*).

Using protein lengths and copy numbers, we calculated the energy costs for each protein per flagellum, and by summing these, obtained the cost for the whole flagellum. The total cost of a single 7.5 μm long (*Turner et al., 2000*) flagellum is $2.32 \times 10^8$ ATP. A large fraction of this cost, 0.99, is in the filament (including the hook). With an average number of 3.4 flagella per *E. coli* cell (*Harshey and Matsuyama, 1994*; *Turner et al., 2000*), the total cost of flagella is $7.88 \times 10^8$ ATP cell$^{-1}$.

Because bacterial flagellum rotation is driven by the flow of protons from one side of the membrane to the other, an estimate of the operating cost requires information on the number of protons crossing the membrane, through the flagellum, per unit time. The energetic costs of this can be expressed in terms of the number of ATP hydrolyses by using the proton/ATP ratio in the ATP synthase, which is 3.33 (*Jiang et al., 2001*). The maximum number of stators, per flagellum, is at least 11 (*Reid et al., 2006*), each of which has two channels (*Braun and Blair, 2001*), and each channel passes 50 protons per revolution (*Gabel and Berg, 2003*; *Samuel and Berg, 1996*), leading to an estimated 1100 protons per revolution, which is similar to the value determined experimentally for *Streptococcus*, 1240 protons per revolution (*Meister et al., 1987*). The *E. coli* flagellum can spin at a maximal rate of 380 Hz (*Chen and Berg, 2000*; *Gabel and Berg, 2003*). The mean swimming speed for *Salmonella*, a close relative of *E. coli*, occurs at a rotation rate of 150 Hz (*Magariyama et al., 2001*). Taking a rotation rate of 200 Hz, a cell division time of 1 hr, 3.4 flagella per cell, and assuming continuous operation we obtain a total operating cost of $1100 \times 200 \times 3.4 \times 3600/3.33 = 8.08 \times 10^8$ ATP cell$^{-1}$.

For many evolutionary considerations, the cost of a structure/function relative to the cost of an entire cell is of interest. For *E. coli*, the construction (or growth) and operating (or maintenance) costs of the cell are estimated to be $1.57 \times 10^{10}$ and $2.13 \times 10^8$ ATP hr$^{-1}$, respectively (*Lynch and Marinov, 2015*), and assuming a cell division time of 1 hr implies a total cell energy cost of $1.59 \times 10^{10}$ ATP. Thus, the costs of constructing and operating flagella relative to the whole-cell energy budget are 5.0% and 5.2%, respectively. The operating cost is close to an earlier estimate of 3.4% but differs from the other estimate of 0.1% (see Introduction). How the 0.1% was obtained is not clear from the source.

Comparing the flagellum operating cost to just the cell *operating* cost leads to an apparent contradiction. The flagellar operating cost exceeds the cell operating cost by 3.8-fold. This mismatch could be due to the variability of *E. coli* cell volume (*Taheri-Araghi et al., 2015*) or the fact that the flagella are not continuously rotating. The cell operating cost may also have been determined under conditions in which cells do not swim. Temperature is probably not an issue as the whole-cell operating costs are normalised to 20°C (*Lynch and Marinov, 2015*), and the flagellum rotation speeds are determined at 23–24°C (*Chen and Berg, 2000*; *Gabel and Berg, 2003*).

## Archaea – cost of the *Pyrococcus furiosus* flagellum

For the archaeal flagellum, we use the reasonably complete structural information on the flagellum of *P. furiosus*, with some additional data from *Sulfolobus acidocaldarius* (*Daum et al., 2017*); see also *Albers and Jarrell, 2018*. The major filament component in *P. furiosus* is FlaB0 (*Nather et al., 2014*), with 1852 copies per μm of flagellum length (*Daum et al., 2017*) (extracted from PDBID: 5O4U). FlaB0 is chemically modified with oligosaccharides (35 sugar units per monomer) (*Daum et al., 2017*; *Fujinami et al., 2014*). We assume that each sugar unit costs as much energy as a single glucose in *E.*

coli, which is 26 ATP (**Mahmoudabadi et al., 2019**). The length of the flagellar filament in *P. furiosus* is stated to be 'a few 100 nm to several µm' (**Daum et al., 2017**). We assume this means 0.3–3 µm, or 1.65 µm on average.

Combining these data yields a construction cost of $2.15 \times 10^7$ ATP per flagellum. As in *E. coli*, a large fraction of this cost is in the filament, 0.98. The number of flagella per *P. furiosus* cell is ~50 (**Daum et al., 2017**), leading to a flagellar cost of $1.07 \times 10^9$ ATP cell$^{-1}$.

In addition to the flagella, *P. furiosus* has a polar cap that organises all flagella into a tuft. The polar cap is rectangular and has a thickness of 3 nm and a linear dimension of 200–600 nm (**Daum et al., 2017**). Assuming a square of $400 \times 400$ nm implies a volume of $4.8 \times 10^{-4}$ µm$^3$. We also determined the total volume of a hexagonal array of protein complexes that is attached to the polar cap. Combining these volumes with the volume per amino acid, we obtain a total polar cap construction cost of $2.43 \times 10^8$ ATP cell$^{-1}$.

The total cell cost is $6.50 \times 10^9$ ATP cell$^{-1}$, which is obtained by applying the *P. furiosus* cell volume (**Daum et al., 2017**) to the regression in **Lynch and Marinov, 2015**, and assuming a 1 hr cell division time. The relative cost for flagella plus polar cap is 20.2% (16.5% for flagella alone).

The rotation of the archaeal flagellum is driven directly by ATP hydrolysis, and 12 ATP are used for each rotation in *Halobacterium salinarum* (**Iwata et al., 2019**). The rotation rate is about 22 Hz (**Iwata et al., 2019**), but this may be an underestimate because in this measurement the flagellum was loaded with a 210 nm diameter bead, and in *E. coli* rotation rates decrease with an increased load (**Gabel and Berg, 2003**). We obtain an energy use of 264 ATP s$^{-1}$ flagellum$^{-1}$. Applying the *H. salinarum* numbers to *P. furiosus*, assuming a 1 hr cell division time (**Nather et al., 2006**), yields an operating cost of $264 \times 50 \times 3600 = 4.75 \times 10^7$ ATP cell$^{-1}$.

*P. furiosus* has a peculiar metabolism and can grow at 100°C (**Kengen, 2017**). We have not considered the impact of these differences on flagellar costs, but this should be investigated in the future.

## Eukaryotes – cost of the *Chlamydomonas reinhardtii* flagellum

For the eukaryotic flagellum, we focus on *C. reinhardtii*, as its flagellum is very well characterised structurally. We include the axoneme, the membrane, and the IFT system. The axoneme contains a central pair of microtubules surrounded by nine doublet microtubules. Each microtubule doublet consists of α- and β-tubulins and about 30 additional proteins (**Ma et al., 2019**). Bound to the outside of each doublet are the radial spokes (**Pigino et al., 2011**), the radial spoke stub (**Barber et al., 2012**), the IC/LC complex (**Heuser et al., 2012**), the CSC (**Dymek and Smith, 2007**; **Pigino et al., 2011**), the Nexin-DRC (**Bower et al., 2013**), the MIA complex (**Yamamoto et al., 2013**), the tether (**Heuser et al., 2012**), the outer dynein arms (**King, 2016**; **King and Patel-King, 2015**; **Ma et al., 2019**), and finally the inner dynein arms (**King, 2013**; **King, 2016**). The central pair of microtubules consist of α- and β-tubulins, and are bound by several other protein complexes (**Carbajal-González et al., 2013**). To obtain the cost of the membrane, we assume a cylinder with a diameter of 0.25 µm (**Khan and Scholey, 2018**) and use *Equation 2*, with $C_L = 406$ ATP (**Lynch and Marinov, 2017**). Finally, we include the construction cost of 279 IFT complexes per flagellum (**Vannuccini et al., 2016**). Combining axoneme, membrane, and IFT, and assuming two flagella of 11 µm in length (**Tuxhorn et al., 1998**), we obtain a *Chlamydomonas* flagellar construction cost of $6.15 \times 10^{10}$ ATP cell$^{-1}$.

The operating cost of the *Chlamydomonas* flagellum determined experimentally on a naked axoneme (flagellum without the membrane) is $8.8 \times 10^4$ ATP s$^{-1}$ µm$^{-1}$ (**Chen et al., 2015**). This is similar to estimates for a generic eukaryotic flagellum, $6.0 \times 10^4$ ATP s$^{-1}$ µm$^{-1}$ (**Raven and Richardson, 1984**); *Paramecium caudatum*, $5.6 \times 10^4$ ATP s$^{-1}$ µm$^{-1}$ (**Katsu-Kimura et al., 2009**) (Methods), assuming a total flagellum length of $1.70 \times 10^5$ µm (*Figure 1—source data 1*); and sea urchin sperm, $6.2 \times 10^4$ ATP s$^{-1}$ µm$^{-1}$ (**Chen et al., 2015**). The *Chlamydomonas* operating cost for a full 9.15 hr cell cycle (**Lynch and Marinov, 2015**) is $6.39 \times 10^{10}$ ATP.

The *Chlamydomonas* whole-cell construction and operating cost, at a 9.15 hr cell division time, is $5.32 \times 10^{12}$ ATP cell$^{-1}$ (**Lynch and Marinov, 2015**), leading to the relative costs of constructing and operating both flagella of 1.4% and 1.2%, respectively. Comparing the flagellar operating cost ($6.39 \times 10^{10}$ ATP) to just the cell operating cost ($2.4 \times 10^{11}$ ATP per cell cycle) reveals that, unlike for *E. coli*, *Chlamydomonas* has enough energy in its operating budget to swim continuously.

## Flagellar construction costs across the tree of life

Flagellar construction costs for a wide range of bacterial and eukaryotic species were obtained by combining the just determined flagellar construction costs of *E. coli* and *Chlamydomonas* with flagellum lengths and numbers, and cell volumes for 196 species (27 bacteria and 169 eukaryotes) (Methods).

For bacteria, we assume that all flagella are built from FliCs of the same size as those in *E. coli* (FliC, 498 amino acids), the same holds for the other flagellar constituents. The absolute flagellar construction cost for bacteria spans 3.6 orders of magnitude, from $1.2 \times 10^8$–$4.4 \times 10^{11}$ ATP (*Figure 1A*), with a median of $7.3 \times 10^8$ ATP. The relative flagellar construction cost spans 2.3 orders of magnitude, from 0.43% to 96% (*Figure 1B*), with a median of 3.7%. The absolute cost of flagella increases with bacterial cell volume, but there is no discernible pattern relating relative costs and cell volume. A power law fit to the bacterial absolute cost vs. cell volume data yields $c_{abs} = 1.1 \times 10^9 V^{0.81 \pm 0.14}$ (exponent ± SE) (with cell volume in µm³ and absolute cost in number of ATP hydrolyses).

For most of the eukaryotes, we assume that flagella are of the same type as that of *Chlamydomonas*. For three phylogenetic groups, kinetoplastids, euglenids, and dinoflagellates, we also include the extra rod that they carry in their flagella besides the axoneme (*Hyams, 1982*; *Maruyama, 1982*; *Portman and Gull, 2010*; *Saito et al., 2003*), which increases the cost per unit length by twofold. We also assume, for all eukaryotes, that the flagellum maintains the same cross-sectional area throughout its length (though see Methods).

The absolute flagellar construction cost for eukaryotes spans 5.3 orders of magnitude, from $2.2 \times 10^{10}$–$4.2 \times 10^{15}$ ATP (*Figure 1A*), with a median of $1.8 \times 10^{11}$ ATP. The relative flagellar construction cost spans 3.1 orders of magnitude, from 0.037% to 44% (*Figure 1B*), with a median of 3.0%. Eukaryotic flagellates (euglenids, kinetoplastids, and other eukaryotes), carrying a small number of flagella, have lower absolute flagellum costs than the hyperflagellated ciliates and parabasalids. This is true even where cell volumes overlap. A power law fit to the eukaryotic flagellate absolute cost data has an exponent of 0.31±0.03, and a joint fit to the ciliates and parabasalids has an exponent of 0.86±0.10. In the plot of relative construction cost against cell volume, eukaryotic flagellates are again clearly separate from the ciliates and parabasalids. Here, the eukaryotic flagellates follow a power law scaling with an exponent of –0.66±0.03. An overview of eukaryotic flagellar and cellular traits is provided in *Figure 1—figure supplement 1*.

# Discussion

## Flagellar costs and benefits

To help understand why species evolve and maintain flagella, and why different domains of life rely on very different kinds of flagella, trait costs need to be compared to trait benefits. Here, benefits are analysed in three steps, from an increase in swimming speed to nutrient uptake to faster cell growth.

Swimming speeds were obtained from the literature for a subset of species from our cost dataset, revealing that swimming speed increases weakly with cell volume when all species are examined together (power law exponent ± SE = 0.13±0.03) (*Figure 2A*). When focussing on just eukaryotic flagellates, we find a volume-independent swimming speed (examined in more detail in the next section), exponent = 0.02 ± 0.09 (*Figure 2A*). There is, however, considerable noise in the data. Previous work, using a different set of organisms, suggested an exponent of 0.22±0.02 for bacteria and eukaryotes combined, and an exponent of 0.14±0.04 for eukaryotic flagellates (*Lynch and Trickovic, 2020*). Besides increasing with cell volume, the swimming speed also increases with absolute flagellar construction cost (exponent = 0.16 ± 0.03; *Figure 2B*).

Next, we compared bacterial and eukaryotic flagella in their ability to generate swimming speed. To allow for a direct comparison between the two groups, swimming speed is expressed in cell lengths per second per ATP invested, revealing that bacterial and eukaryotic flagella follow a common trend (*Figure 2C*), suggesting that there is no large difference in cost-effectiveness between the two groups once scaling with cell size has been taken into consideration. However, data for each flagellar type (bacterial or eukaryotic) only cover a limited, and mostly non-overlapping, volume range. Whether the common trend will be maintained over a broader range of cell volumes is unclear, but doubtful for the eukaryotic flagellum (see below).

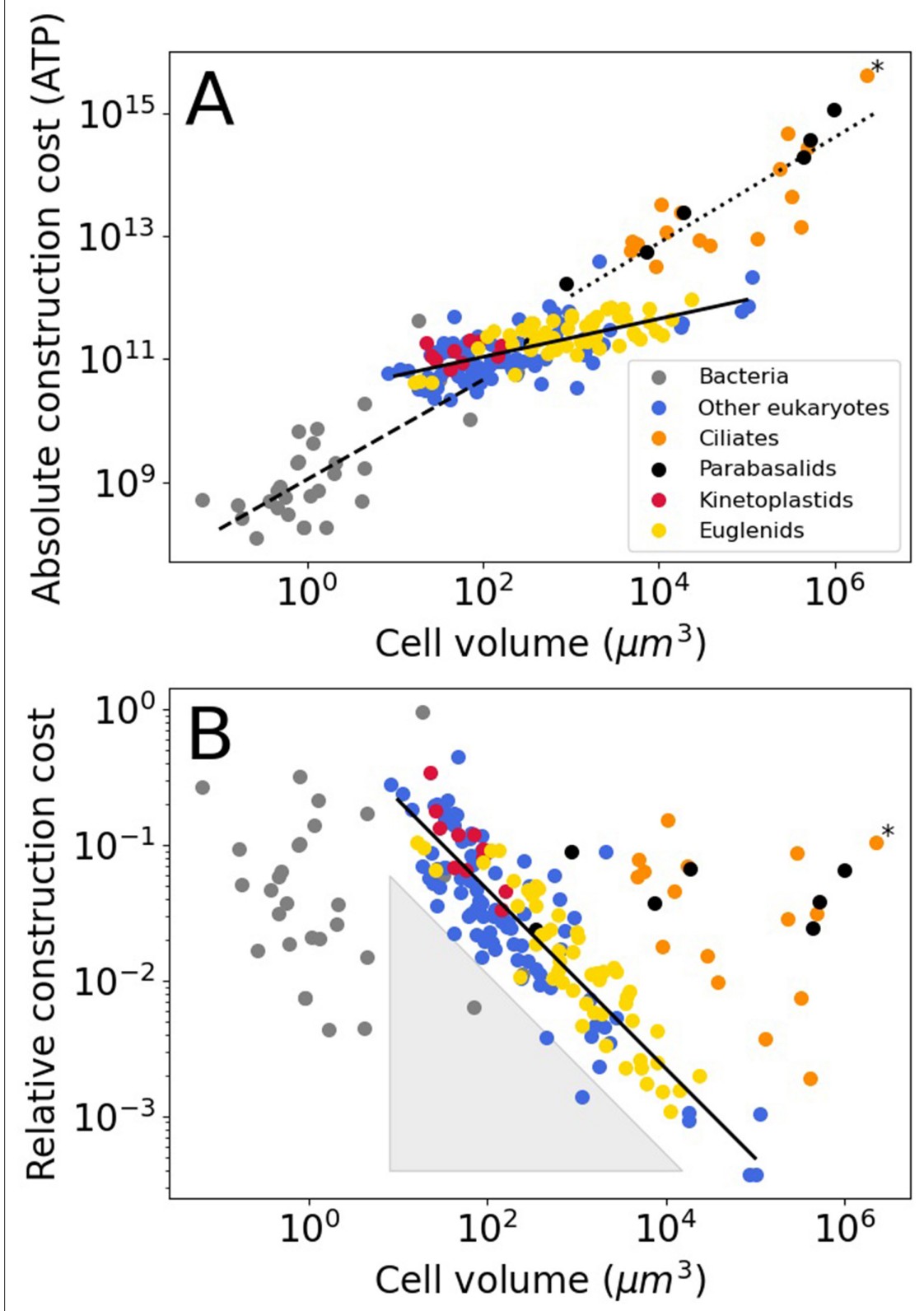

**Figure 1.** Construction costs of flagella in bacteria and eukaryotes as a function of cell volume. Each point denotes a species. The costs are the sum over all flagella present on a single cell. (**A**) The absolute construction cost. The lines are power law fits to bacteria, eukaryotic flagellates (euglenids, kinetoplastids, and 'other eukaryotes'), and ciliates and parabasalids. The equations are: $c_{abs} = 1.1 \times 10^{9} V^{0.81 \pm 0.14}$, $c_{abs} = 2.6 \times 10^{10} V^{0.31 \pm 0.03}$, and $c_{abs} = 2.8 \times 10^{9} V^{0.86 \pm 0.10}$, respectively (exponent ± SE). (**B**) The construction costs of flagella relative to the construction costs of the entire cell. The

*Figure 1 continued on next page*

*Figure 1 continued*

line is a power law fit to the eukaryotic flagellates: $c_{rel} = 0.98V^{-0.66\pm0.03}$. The grey-shaded triangle is explained in the Discussion. In both panels the asterisk marks *Opalina ranarum*, which resembles ciliates in its flagellar distribution but does not belong to the ciliate clade. Data in *Figure 1—source data 1*.

The online version of this article includes the following source data and figure supplement(s) for figure 1:

**Source data 1.** Table of flagellar and cellular properties for bacterial and eukaryotic species.

**Figure supplement 1.** Overview of flagellar and cellular properties of eukaryotic species.

**Figure supplement 1—source data 1.** Table of flagellar numbers and lengths of eukaryotic species.

**Figure supplement 1—source data 2.** Table of cell aspect ratios for eukaryotic species.

**Figure supplement 2.** Relative flagellar construction costs of special cases plotted against cell volume.

**Figure supplement 2—source data 1.** Table of alternative flagellar construction costs.

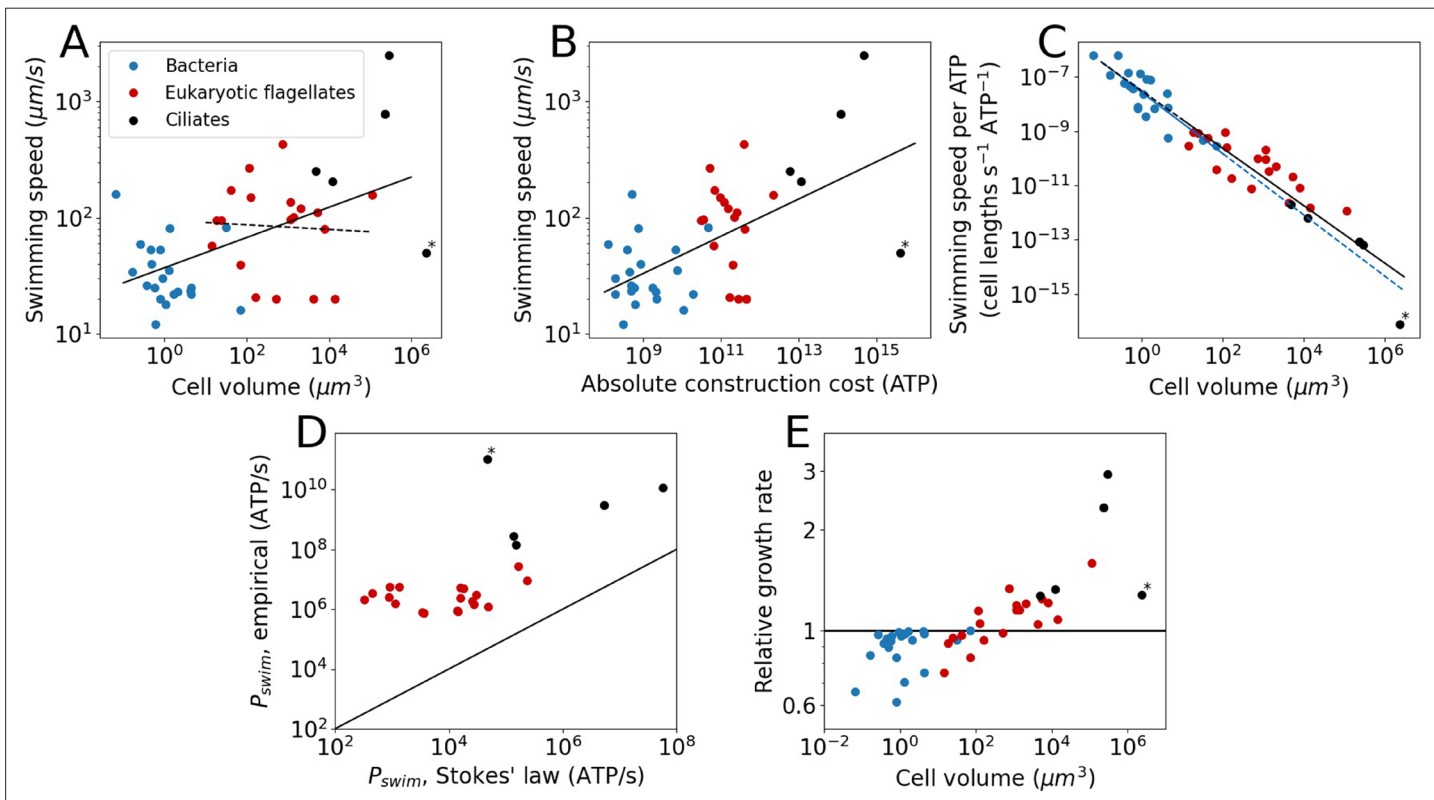

**Figure 2.** The cost and benefits of flagellar motility. Each point denotes a species. (**A**) Swimming speed plotted against cell volume. Plotted are all species from the cost dataset for which the swimming speed is known. This includes bacteria, eukaryotic flagellates, and ciliates. The continuous line is the fit to all species: $v = 37V^{0.13\pm0.03}$ (exponent ± SE). The dashed line is a fit to the eukaryotic flagellates: $v = 95V^{-0.02\pm0.09}$. The legend holds for the entire *Figure 2*. (**B**) Swimming speed plotted against the absolute flagellar construction cost. The line is the fit: $v = 1.2c_{abs}^{0.16\pm0.03}$. (**C**) Swimming speed (in cell lengths s$^{-1}$) per ATP of construction cost plotted against cell volume. The solid blue line is the fit to the bacterial data: $v_{ATP} = 2.7 \times 10^{-8}V^{-1.13\pm0.16}$. The solid black line is the fit to the eukaryotic data (both flagellates and ciliates): $v_{ATP} = 3.1 \times 10^{-8}V^{-1.06\pm0.11}$. The dashed lines are extrapolations. (**D**) Comparison of the swimming power, or operating cost, calculated from Stokes' law with empirical values. This gives an indication of the efficiency of converting chemical energy into swimming power. The line indicates equality. (**E**) The relative growth rate as a function of cell volume for cells in a medium with a homogenous distribution of small molecule nutrients, comparing cells with flagella to cells without flagella. In all panels the asterisk marks *Opalina ranarum*, which resembles ciliates in its flagellar distribution but does not belong to the ciliate clade. Data in *Figure 2—source data 1*.

The online version of this article includes the following source data for figure 2:

**Source data 1.** Table with swimming speeds, swimming power, and cell growth rate.

To swim, a cell not only needs to construct a flagellum, but to operate it as well. We estimate operating costs in two separate ways. The first deploys Stokes' law, which describes the power needed to move a sphere of radius, $r$, at a speed, $v$, through a medium with viscosity, $\eta$:

$$P = 6\pi\eta r v^2 \tag{3}$$

The second approach multiplies the ATP hydrolysis rate per μm of flagellum with the total flagellar length for each species (considering only eukaryotes). For the ATP hydrolysis rate, we used the average of the empirically determined values for *Chlamydomonas*, *Paramecium*, and sea urchin sperm (68,700 ATP s$^{-1}$ μm$^{-1}$; see above). Comparing these two operating costs (***Figure 2D***) reveals that eukaryotes have a mean efficiency of 0.7% for the conversion of chemical energy (ATP) into swimming power, which is similar to earlier reports of swimming efficiencies for single-celled organisms (***Chattopadhyay et al., 2006***; ***Osterman and Vilfan, 2011***; ***Purcell, 1997***).

Faster swimming speeds can be used to increase nutrient uptake rate. The benefit of swimming for nutrient uptake, or the lack thereof, has been discussed by others, but without reference to the flagellar construction cost (***Purcell, 1977***; ***Wan and Jékely, 2021***). However, the net fitness benefit also depends on what it costs to build and operate the flagella that propel a cell. To determine swimming gains, the amount of nutrients that is obtained by swimming needs to be compared to the amount of nutrients that is obtained by a stationary cell through diffusion. For a homogeneous nutrient distribution this is accomplished by calculating the Sherwood number, $Sh$ (***Guasto et al., 2012***):

$$Sh = \frac{1 + \left(1 + 2\frac{vL}{D}\right)^{\frac{1}{3}}}{2}. \tag{4}$$

Here, $v$ is the swimming speed (in μm s$^{-1}$), $L$ is the cell diameter (in μm), and $D = 10^3$ μm$^2$ s$^{-1}$ is the small molecule diffusion coefficient (e.g. amino acids). A stationary cell has an $Sh$ of 1, whereas an $Sh$ of 1.1 means that swimming yields a 10% increase in nutrient uptake rate. For a flagellum to be selectively advantageous, the gain in nutrients must outstrip the cost of the flagellum. If the excess nutrients are used just for increasing the growth rate, then a relative growth rate is obtained as (Methods):

$$Relative\ growth\ rate = \frac{Sh}{1 + c_{rel,c+o}}. \tag{5}$$

Here, $c_{rel,c+o}$ is the relative flagellar cost, which includes both the construction and operating cost. It is assumed that the growth rate is nutrient limited and that for all species the flagellar operating cost is equal to the construction cost (***Table 1***). For species with a cell volume <10$^2$ μm$^3$, flagella constitute a net loss in nutrients and decrease in growth rate (***Figure 2E***). This suggest that for these species, flagella cannot profitably be used for gathering nutrients in a homogenous medium. Note that this conclusion is unaffected by the possible overestimation of the bacterial flagellar operating cost noted in the results section. A complete removal of operating costs for the bacteria in ***Equation 5***, leaving only construction costs, still results in bacteria being mostly unable to maintain flagella in a homogenous environment. For species with a cell volume >10$^3$ μm$^3$, flagella can provide a net gain in nutrients and therefore constitute an increase in growth rate.

## Scaling properties of flagellar energy costs and swimming speed

Here, we compare the empirical power-law relationships between flagellum construction cost, swimming speed, and cell volume (***Figures 1B and 2A***) with expected relationships derived from simple physical principles. If a simple model captures the general observed behaviour, considerations of morphological details and ecological backgrounds of each species and cells of different volumes become secondary issues. From Stokes' law, already described above (***Equation 3***), we derived two power laws to compare with the empirical patterns in ***Figures 1B and 2A***; detailed derivations can be found in the Methods.

The first power law concerns the relation between cell volume and relative flagellar construction cost observed for the eukaryotic flagellates in ***Figure 1B***. Stokes' law describes the power required for swimming, that is, the flagellar *operating* cost. To make the connection to the flagellar *construction* cost in ***Figure 1B***, it is assumed that flagellar operating cost and construction cost are linearly related (Methods). We also assume a spherical cell shape. A relative flagellar construction cost is obtained by dividing the absolute flagellar cost by the whole-cell cost which is obtained from the cell volume using an empirical relation (***Lynch and Marinov, 2015***). Finally, based on the swimming speed data

for eukaryotic flagellates (**Figure 2A**), we assume that swimming speed is independent of cell volume. These considerations yield (Methods):

$$c_{rel} = a_1 V^{-0.64 \pm 0.04}. \tag{6}$$

Here, $c_{rel}$ is the relative flagellar construction cost, $a_1$ is a constant, and $V$ is the cell volume. The exponent in **Equation 6** matches with the empirically determined exponent for eukaryotic flagellates (which excludes ciliates and parabasalids), $-0.66 \pm 0.03$ (**Figure 1B**). This suggests that the scaling of flagellar construction cost is determined by simple physical principles combined with the fact that swimming speeds do not vary with cell volume. The constant swimming speed is presumably the result of ecological factors. This result can be extended to any set of species in **Figure 1B** that are arranged along a diagonal with a slope of $-0.64$, so that, for example, all species along the bottom left edge of the ciliate distribution should also have the same swimming speed. Another corollary of this result is that there appears to be a (soft) lower limit to the swimming speed of eukaryotic flagellates, as there is a distinct lack of eukaryotic species in the grey shaded triangular area shown in **Figure 1B**.

The second power law concerns the relation between the cell volume and the swimming speed that is observed in **Figure 2A** (solid line). Here, unlike the previous paragraph, we consider not just the eukaryotic flagellates, but the bacteria and ciliates as well. As above, we start from Stokes' law and assume a spherical cell shape. We also assume that the operating cost is a fixed fraction of the total cell operating (or maintenance) cost. These considerations yield (Methods):

$$v = a_2 V^{0.27 \pm 0.04}, \tag{7}$$

where $v$ is the swimming speed and $a_2$ is a constant. The model exponent in **Equation 7** is somewhat higher than the empirical result (**Figure 2A**; $0.13 \pm 0.03$), as well as that from a previous analysis (**Lynch and Trickovic, 2020**), $0.22 \pm 0.02$. This discrepancy may be due to the decreased relative energy investment in flagella in eukaryotic flagellates of larger cell volume (**Figure 1B**).

## Proteins from different parts of a flagellum experience different evolutionary forces

Flagella are composed of many different proteins which are present in a broad range of copy numbers. For example, a single *E. coli* flagellum contains 1 FliJ protein (part of exporter), 10 FliD's (filament cap), 100 FlgE's (hook), and $10^4$ FliCs (filament). Due to the difference in protein copy number, the addition of a single amino acid to FliC has a larger impact on the cell energy budget than a single amino acid addition to FliJ, and thus requires a higher positive contribution to the cell fitness to be favoured by natural selection. Also, in the absence of any positive effect on fitness, a single amino acid addition to FliC is less likely to drift to fixation.

Here, we quantify the relative cost of adding a single (average) amino acid to selected flagellar proteins of all species for which we have data. For bacteria, the relative energy cost of adding an (average) amino acid to flagellar protein in bacterial species $j$ is:

$$c_{relBac} = \frac{c_{AA} N_{copy,i} N_{flag,j}}{2.7 \times 10^{10} V_j^{0.97}}, \tag{9}$$

where $c_{AA}$ is the average amino acid cost 29 ATP; (**Lynch and Marinov, 2015**), $N_{copy,i}$ is the protein copy number in a flagellum, $N_{flag,j}$ is the number of flagella, and $V_j$ is the cell volume. The denominator is the total cell construction cost growth cost; (**Lynch and Marinov, 2015**). For eukaryotes, we use a slightly different expression:

$$c_{relEuk} = \frac{c_{AA} N_{1\mu m,i} L_{flag,j}}{2.7 \times 10^{10} V_j^{0.97}}, \tag{10}$$

where $N_{1\mu m,i}$ is the protein copy number per μm of flagellum, and $L_{flag,j}$ is the total flagellum length (the sum over the lengths of all flagella of a single cell). **Equations 9 and 10** are different because most of the bacterial flagellar proteins are present only at the flagellar base, and thus do not increase their copy number with flagellar length, whereas for eukaryotic flagella all proteins are present throughout the length of the flagellum so that their copy number tracks the flagellar length. The resultant cost distributions (over all species) for selected flagellar proteins are shown in **Figure 3**.

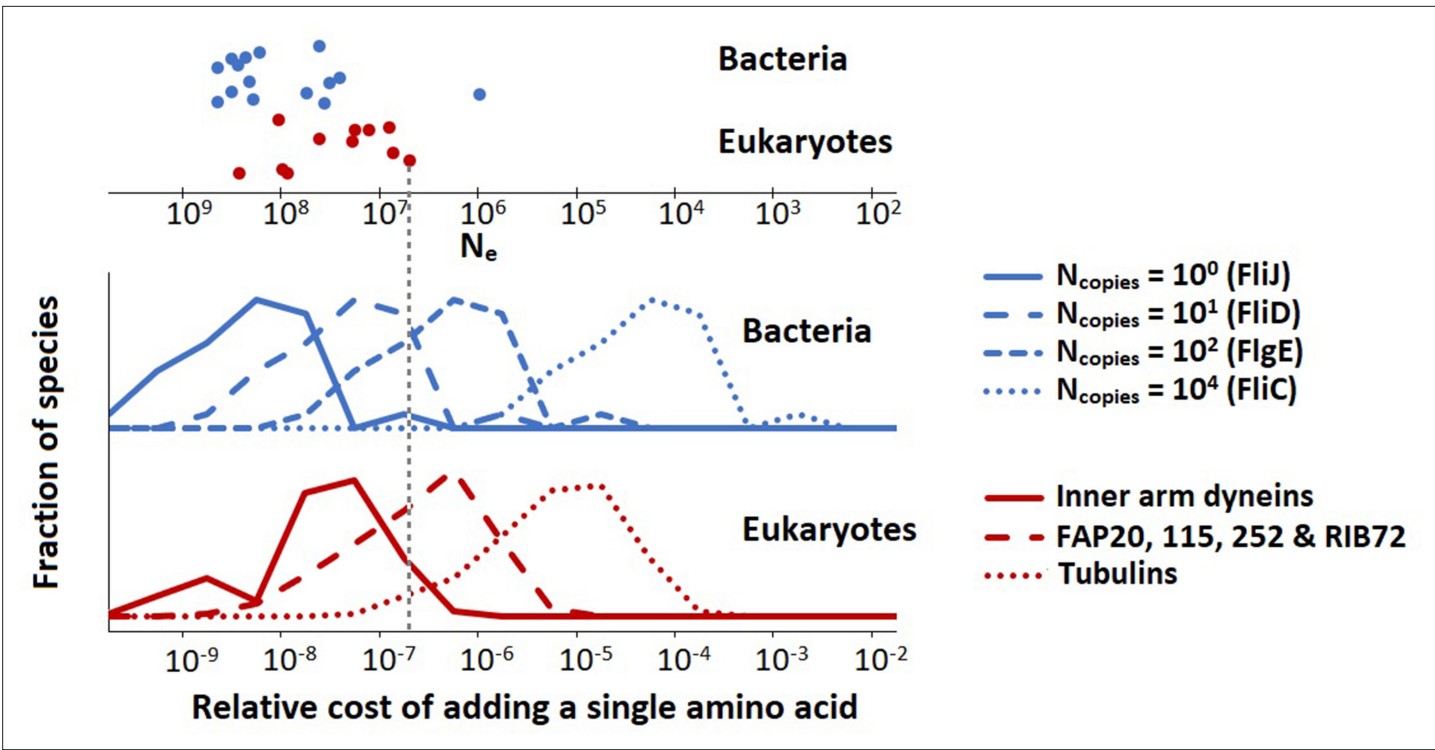

**Figure 3.** The population-genetic environment of different flagellar proteins for bacteria and eukaryotes. The different distributions are for different proteins, with varying copy numbers, within the same flagellum. The spread within each distribution reflects the variation of relative flagellar construction costs over all bacterial or eukaryotic species in our construction cost dataset. The points on the top show the effective population sizes, $N_e$, of different species of bacteria and eukaryotes (**Lynch and Trickovic, 2020**) (the vertical spread of the datapoints is for visualisation). The bacterial flagellar protein names were taken from *Escherichia coli*. Inner arm dyneins are present in 94 copies per μm of flagellum, for FAP20, etc., this number is 1125, and for tubulins it is 29,125. The vertical dashed line is explained in the main text. Data in **Figure 3—source data 1**.

The online version of this article includes the following source data for figure 3:

**Source data 1.** Table with relative costs of adding an amino acid to flagellar proteins in various species.

These data can be used to determine the minimal fitness benefit that an amino acid addition needs to bestow upon the cell for the net fitness effect to be positive. We assume that for the small energy costs considered here, the relative energy cost is equal to the baseline loss in fitness so that the fitness before the amino acid addition is scaled to 1 and after the addition is $1 - c_{relBac}$ or $c_{relEuk}$ (**Ilker and Hinczewski, 2019**; **Lynch and Marinov, 2015**; **Lynch and Trickovic, 2020**). If the amino acid is added to FliC, the $c_{relBac}$ is roughly between $10^{-6}$ and $10^{-3}$ (depending on the species). The beneficial effect of the amino acid addition needs to exceed this value by an amount exceeding the power of random genetic drift to be promoted by positive selection; and similarly, in the absence of any ecological fitness advantage, the insertion will be vulnerable to fixation by effectively neutral processes should the power of genetic drift (defined as the inverse of the effective population size, $N_e$) exceed the cited values. For the low copy number FliJ, the comparable benchmarks ($c_{relBac}$) are between $10^{-10}$ and $10^{-7}$, so the vulnerability to drift is increased by several orders of magnitude. The key point is that low copy number flagellar proteins are expected to be more variable in length than those in high copy number, especially in species with small effective population sizes. This point is illustrated in *Figure 3* with the vertical dashed line (indicating a species with an effective population size, $N_e$, of $5 \times 10^6$). Proteins with copy numbers that put them to the left of this line can accumulate amino acids neutrally, whereas the higher copy number proteins to the right of this line cannot. Similar considerations hold also for amino acid substitutions.

## Evolution of the eukaryotic flagellum
Archaea, bacteria, and eukaryotes each have their own type of flagellum. The reason for this independent evolution is an unsolved problem in evolutionary cell biology, made perhaps more important by

the fact that the transition to a eukaryotic flagellum is part of the plethora of cellular modifications that led to eukaryotic cells. The three flagellar types are so different, in both assembly and thrust generating mechanisms, that it is hard to imagine a gradual transition between any of them. However, given that many prokaryotic and eukaryotic species do not possess a flagellum at all, the different flagellar types likely evolved de novo, in lineages initially devoid of flagella. A eukaryotic flagellum might also have evolved alongside a prokaryotic one, initially carrying out a separate function such as gliding or sensing. There are in fact bacterial species such as *Vibrio parahaemolyticus*, with two kinds of flagella, a thick one for swimming and multiple thin ones for gliding (both of the bacterial type) (**McCarter, 2004**).

For the many prokaryotic species lacking flagella, there may typically be enough food in the vicinity so that the costs of a flagellum outweigh its benefits, resulting in its loss (**Mitchell et al., 1995**). This may have been true also for the prokaryotic lineage that led to the eukaryotes. If so, the unique structure and mechanism of the eukaryotic flagellum may have owed more to contingency than to an advantage in swimming ability. However, eukaryotic cells also tend to be larger in volume than prokaryotic cells, and it is worth considering the consequences of this for the choice of flagellar type. For instance, it may not be possible for the thin prokaryotic flagellum to provide the thrust required to propel a large cell, as the flagellum would buckle under the load. This points to a possible advantage of possessing a thicker, and thereby stronger, eukaryotic flagellum. However, a few observations argue against this possibility. First, multiple bacterial flagella can be combined into a single large flagellum, as is the case for the magnetotactic bacterium MO-1 (**Ruan et al., 2012**). Second, a large bacterial cell could simply have many small flagella, as is the case for the giant bacterium *Epulopiscium* (**Clements and Bullivant, 1991**). Finally, there is a large single-celled eukaryotic species whose swimming is known to be powered by bacterial flagella, *Mixotricha paradoxa*, accomplished by recruiting bacterial species as motility symbionts (**Wenzel et al., 2003**). It remains to be determined whether the prokaryotic flagellum-powered motility of *Epulopiscium* and *M. paradoxa* can compete with the eukaryotic flagellum-powered motility of large eukaryotes. Combining flagellum construction costs with swimming speeds for a diversity of bacterial and eukaryotic species with a range of cell volumes (**Figure 2C**) suggests that there is no large advantage to having a eukaryotic flagellum in terms of the cost-effectiveness of generating swimming speed. However, a minor advantage cannot be ruled out. The absence of an obvious advantage for the eukaryotic flagellum lends credence to the view that it originally evolved in a lineage lacking a flagellum, absolving it from the requirement of outperforming its predecessor. However, it may be that the advantage of the eukaryotic flagellum lay not in its ability

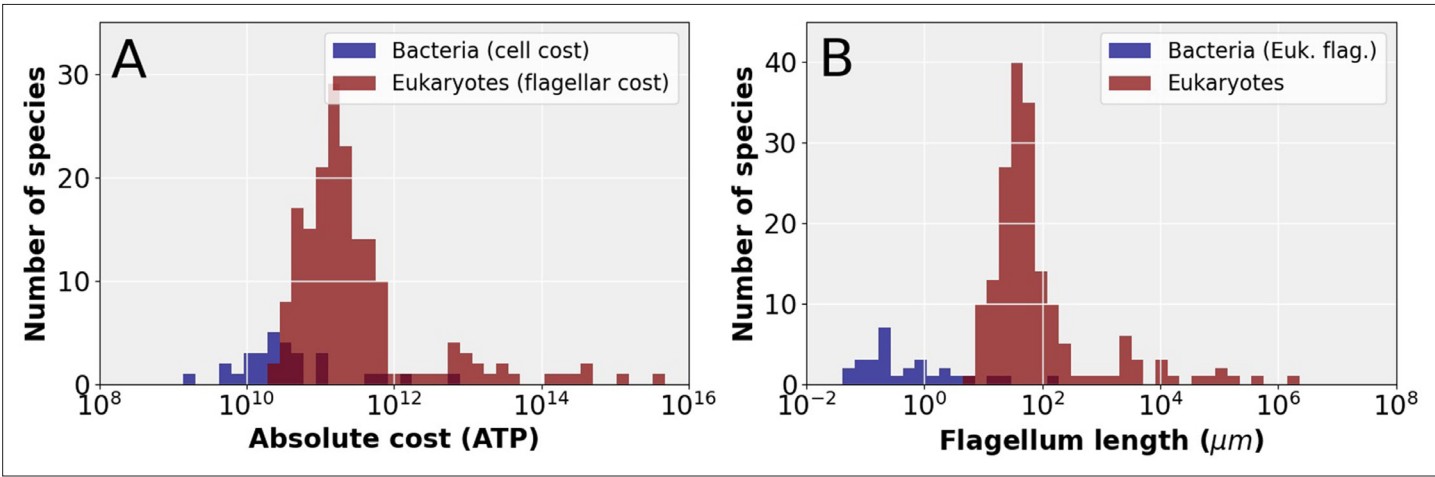

**Figure 4.** The cost of eukaryotic flagella compared to bacterial cell and flagellar budgets. (**A**) Histograms comparing the total cellular cost (excluding maintenance) for bacterial species to the cost of constructing flagella in eukaryotic species. (**B**) Histograms comparing the total *eukaryotic* flagellum length of bacteria and eukaryotes, where for the bacteria we have taken the absolute flagellar construction cost for each bacterial species and divided that by the per µm construction cost of the eukaryotic flagellum, to obtain the hypothetical length of the eukaryotic flagellum affordable to each bacterial species. Data in **Figure 4—source data 1**.

The online version of this article includes the following source data for figure 4:

**Source data 1.** Table with flagellum lengths, flagellum construction costs, and cell construction costs.

to generate speed but in its ability to generate a variety of beat patterns, giving it enhanced agility. A counterpoint is that the investment in a single eukaryotic flagellum could pay for multiple prokaryotic flagella, which together may provide the same level of agility. The data required for a systematic comparison of the agility bestowed upon cells by prokaryotic and eukaryotic flagella is currently lacking.

Next, we consider the question: does the small cell volume of prokaryotic organisms prevent the acquisition of a eukaryotic flagellum? To answer this question, we first compare the absolute construction cost of entire bacterial cells to the construction cost of eukaryotic flagella (*Figure 4A*), revealing that the cheapest eukaryotic flagella are in fact more expensive than the entire cells of many bacterial species. Next, we examine the observed absolute flagellar investment in these bacterial species and ask what length of *eukaryotic* flagellum they could afford. These lengths are compared with the total flagellum length of eukaryotic species, revealing that most bacterial species cannot afford full length eukaryotic flagella (*Figure 4B*). Two observations suggest that the short and stumpy eukaryotic flagella these bacteria could afford would be highly ineffective. First, all eukaryotic flagella that have been observed, appear to be long and slender. Second, all small eukaryotes have a high relative investment in flagella (*Figure 1B*), suggesting that short (but cheaper) flagella are not up to the task. Reducing the cost of the eukaryotic flagellum by decreasing the number of microtubule doublets occurs in some species (*Prensier et al., 1980*; *Schrevel and Besse, 1975*), but its rarity suggests that it is associated with a considerable reduction in swimming ability.

The possibility of an evolutionary origin of a reduced form of the eukaryotic flagellum in small cells cannot be ruled out. Consider the small alga *Micromonas pusilla* which has a cell volume of about 1 µm$^3$ and a eukaryotic flagellum (*Simon et al., 2017*). Here, however, unlike most eukaryotic flagella, the outer microtubule doublets and motor proteins are only present in the proximal 10–20% of the flagellar length. The remainder of the flagellum consists of the inner pair microtubules surrounded by a membrane (*Vaulot et al., 2008*). Thus, the basic machinery that operates the eukaryotic flagellum, for example, outer doublet microtubules, dynein motors, and radial spokes, may have evolved in a small prokaryote-sized cell; but the common eukaryotic flagellum, with this machinery present all along its length, could only evolve in larger cells.

## Conclusion

Using available structural data, we have determined the energy cost of building flagella in ~200 unicellular species, including archaea, bacteria, and eukaryotes, recorded the swimming speeds for a subset of these, and reached the following conclusions. There appears to be a minimum swimming speed for eukaryotic flagellates, independent of cell volume. Flagella in small cells cannot be maintained solely for the collection of nutrients in homogenous media. Proteins from the same flagella but with lower copy numbers may experience a more permissive evolutionary environment, being able to accumulate amino acids neutrally and requiring lower beneficial effects for amino acid additions to be favoured by selection. Eukaryotic and prokaryotic flagella follow a common trend in the cost-effectiveness of generating swimming speed. Finally, the eukaryotic flagellum is only energetically affordable for cells that are larger than the typical prokaryote.

# Materials and methods
## Data extraction

To calculate absolute and relative construction costs of flagella over a whole range of species, cell volumes and flagellar parameters were extracted from the literature. The cell volume was calculated from cell lengths (always excluding the flagellum), and widths, and occasionally also the cell depth. For bacteria a spherocylindrical cell shape and for eukaryotes a spheroidal cell shape was assumed. For some species, numbers for cell lengths, widths, and depths were not reported, so we extracted these from microscopy images, or, in some cases, from drawings.

The bacterial flagellum is helical but when a length is reported this could mean the arclength of the helix or the base to end distance. Wherever possible we used or calculated the arclength of the helix. When it was not clear what the reported length referred to, we assumed it was the arclength.

For ciliates and parabasalids, the number and length of flagella were in most cases not reported, so they were extracted from microscopy images and drawings. We made use of the regular spacing

between flagella and the fact that they are arranged in rows and assumed that dikinetids have two flagella. Oral flagella were in most cases not included.

In 10 cases there were more than one, and somewhat differing, datasets for the same species. These were included as separate points.

## Costs of protein turnover

The energetic costs of protein degradation and resynthesis (protein turnover) would show up in the whole-cell operating cost. Since the cell operating cost is typically a lot lower than the whole-cell construction cost (*Lynch and Marinov, 2015*), and because we don't know the exact flagellar protein turnover cost, we did not include the turnover costs in our flagellar cost calculations. Another argument in support of this choice is that the flagella are enormous protein complexes with exchange of subunits mainly happening at the tip, likely increasing the stability of flagellar proteins against proteolysis. One additional source of protein is shearing and resynthesis of flagella. However, in the absence of data on the number of shearing events per cell cycle, the contribution of this to flagellar protein turnover could not be determined.

Another kind of the protein turnover, relevant mainly for the eukaryotic flagellum, is the exchange of protein subunits at the flagellar tip. The exchange of protein subunits at the flagellar tip depends on a pool of subunits that is maintained by the IFT system. As such the cost of the axoneme tip dynamics is constituted mostly of IFT transport costs (*Bauer et al., 2021*). The approximate operating cost of IFT can be obtained from motor protein activity (with dynein and kinesin moving cargo in opposite directions). The number of IFT particles arriving at the flagellar tip is 1.25 per sec (*Bauer et al., 2021*), the number of dynein motors per IFT particle is 13 (*Webb et al., 2020*), and the step size of a dynein motor is 8 nm (*Kinoshita et al., 2018*). If we assume 1 ATP hydrolysis per dynein step, a 11 μm long flagellum, a cell cycle time of 9 hr, and that the use of kinesin costs the same as dynein, we obtain the total cost of transport: $1.25 \times 13 \times 11/0.008 \times 1 \times 9 \times 3600 \times 2 = \sim 1.4 \times 10^9$ ATP, or almost 5% of total flagellar cost. A tubulin dimer also uses up an ATP each binding cycle, but this is a small contribution to the total ATP cost given that each tubulin dimer has a construction cost of ~25,000 ATP. For costs of tubulin construction to balance tubulin exchange, (an equivalent of) the entire flagellum needs to be dissociated and reassembled 25,000 times, which is unlikely. Overall, protein turnover contributes little to flagellar costs.

## Calculating the average volume per amino acid

The average molecular weight per amino acid residue is 110 g mol$^{-1}$. The partial specific volume of proteins is 0.73 mL g$^{-1}$ or $0.73 \times 10^{-6}$ m$^3$ g$^{-1}$ (BNID 104272 and 110540) (*Milo et al., 2010*). Multiplying the amino acid molecular weight with the partial specific volume yields $80.3 \times 10^{-6}$ m$^3$ mol$^{-1}$. Dividing this by Avogadro's number gives $1.33 \times 10^{-28}$ m$^3$ residue$^{-1}$ or $1.33 \times 10^{-10}$ μm$^3$ residue$^{-1}$.

## Operating cost of *P. caudatum* flagellum

The oxygen consumption rate of *P. caudatum* at different swimming speeds has been reported in the literature (*Katsu-Kimura et al., 2009*). This led to an estimate of the energy use for swimming at 1 mm s$^{-1}$, which is $2.84 \times 10^{-6}$ J hr$^{-1}$. The hydrolysis of ATP yields $5.0 \times 10^4$ J mol$^{-1}$ (*Milo and Phillips, 2016*). From these numbers, and the fact that *P. caudatum* has $1.70 \times 10^5$ μm worth of flagellum, operating cost per μm of flagellum length was determined: $5.60 \times 10^4$ ATP s$^{-1}$ μm$^{-1}$.

## Flagellum assumptions

For the eukaryotic species that do not belong to the euglenids, kinetoplastids, or dinoflagellates, we treat the flagellum as if it were a *Chlamydomonas* flagellum, but with differences in length. This flagellum has a constant thickness throughout its length. For species that do belong to the euglenids, kinetoplastids, or dinoflagellates, the cost per unit length of flagellum was doubled to account for the presence of a rod. Here, we have also assumed constant thickness of the flagellum along its length. The constant thickness assumption appears to be borne out by the rod bearing *Trypanosoma brucei* (*Portman and Gull, 2010*), *Peranema trichophorum* (*Saito et al., 2003*), and *Ceratium tripos* (*Maruyama, 1982*) (*Ceratium tripos* is not in our dataset but *Ceratium fusus* is). We did find four species among the eukaryotic flagellates that have a flagellum that starts broad at the base and becomes thinner along its length, and that have a detailed enough description for us to make a cost

calculation based on volume (assuming the same cost density as the *Chlamydomonas* flagellum). These species are *Anaeramoeba ignava*, *Anaeramoeba gargantua* (*Táborský et al., 2017*), *Dinematomonas valida* (*Larsen and Patterson, 1990*), and *Psammosa pacifica* (*Okamoto et al., 2012*). In *Figure 1—figure supplement 2* we plot the newly calculated relative cost as a function of the cell volume, as well as the old cost for comparison. This shows that there may be more overlap between the eukaryotic flagellates, and ciliates and parabasalids. Besides these four there appear to be more cells with somewhat thicker flagella at their base or throughout their length based on the drawings in *Larsen and Patterson, 1990*; *Patterson and Simpson, 1996*, but not enough detail is given for us to perform any calculations.

Many eukaryotic flagella have hairs, either many along their length (mastigoneme) or one at the tip (acroneme), or vanes (*Janouškovec et al., 2017*; *Tikhonenkov et al., 2014*). As we do not know the detailed composition or exact size of these, we did not include their costs.

In calculating the relative costs of the flagella with respect to the total cell budget we have assumed that the flagella are expressed in every generation. If the flagella were expressed only once per 10 generations, then the relative cost, in evolutionary terms, would be 10-fold lower.

Detailed structural information is only available for a limited number of eukaryotes. In principle, there could be differences in the protein composition of the axoneme between different groups of organisms. A comparison of the *Chlamydomonas* flagellum with that of the ciliate *Tetrahymena* revealed only minor differences in axoneme structure (*Pigino et al., 2012*), which are inconsequential for our cost calculations. Some gametes are known to have 6+0 or 3+0 axoneme structure (*Prensier et al., 1980*; *Schrevel and Besse, 1975*), but these structures appear to be rare.

In our analysis of flagellar construction cost and swimming speeds we have assumed that all flagella were used for swimming. Flagella can also be used for gliding (*Heintzelman, 2006*) or capturing food. With more data on flagellar arrangements and descriptions of how flagella are in fact used, we will, in the future, be able to tease these different situations apart.

## Completeness of the eukaryotic flagellum construction costs

The eukaryotic flagellum has been subject to proteomics studies that have attempted to identify the full complement of flagellar proteins (*Pazour et al., 2005*; *Yano et al., 2013*). Unfortunately for our purposes, these studies did not yield quantitative information on the abundance of flagellar proteins. Therefore, we turned instead to structural data for our construction cost calculations. This structural data, however, may be incomplete (though we have no reason to believe that it is) and does not include freely diffusing proteins that are present in, and contribute to the function of, the flagellum. To assess completeness, we turn to an estimation of volume occupancy of flagellar proteins. Adding up the volumes of all proteins in the axoneme, the IFT particles, and the membrane protein domains that we assume point inwards, we come to a total protein volume fraction of 0.18. For the flagellum internal volume, we used a flagellum length of 11 µm and a width of 0.24 µm (the width excludes the membrane). This volume fraction is higher than the macromolecular volume fraction of the *E. coli* cytoplasm, 0.16 (*Konopka et al., 2009*). Adding a lot of extra protein on top of the volume occupancy of 0.18 would likely cause severe crowding effects that interfere with the functioning of IFT and the dynein motor activity that bends the flagellum. As such we expect that our construction costs are close to complete.

For the archaeal flagellum we have included posttranslational modifications, which account for about 17% of total costs. In the eukaryotic flagellum there are posttranslational modifications to tubulin. The exact amount of biomass contributed is unclear, but the following considerations suggest that these modifications would increase the construction cost estimates by only a small amount. First, the main contributor to the posttranslationally added biomass appears to be a polyglutamylation of between 1 and 20 residues per tubulin (*Janke et al., 2008*), which constitutes a smaller cost than the 35 sugar groups added to the archaellin protein in the archaeal flagellum. Second, the tubulin monomer is larger (~450 AAs) than the archaellin (207 AA), reducing the relative cost of the modification. Third, not all tubulin monomers are modified to the same degree (the A tubule and the central pair tubules exhibit a much lower level of polyglutamylation than the B tubule) (*Orbach and Howard, 2019*; *Wloga et al., 2017*). Fourth, the contribution of tubulin to total eukaryotic flagellum cost (~30%) is smaller than that of archaellin to the archaeal flagellum cost (~98%). Together these considerations suggest a cost of posttranslational modifications that is ~50-fold lower than the

posttranslational modifications in the archaeal flagellum, contributing <1% to eukaryotic flagellum costs.

To determine the cost of the protein complexes associated with the central pair microtubules in the eukaryotic flagellum, we used lower resolution structural data in which the individual proteins were not resolved (*Carbajal-González et al., 2013*). During the review process two papers with high-resolution structures of the central apparatus (central pair microtubules + associated protein complexes) were published (*Gui et al., 2022*; *Han et al., 2022*). To verify our estimate of the central apparatus cost we calculated the fraction of the mass of the central apparatus that is contributed by the central pair-associated protein complexes (excluding the tubulin), which yields ~69%. This is similar to the ~71% determined from the higher resolution structure of the central apparatus (*Han et al., 2022*). Because the tubulin number in the central pair is the same in both cases (624 tubulins per 96 nm repeat), and because the cost is proportional to the mass, our estimate of the central apparatus cost is validated by the higher resolution structural information.

## Derivation of the relative growth rate equation

Here, we derive the relation of the relative growth rate to the Sherwood number, $Sh$, and the relative flagellar construction and operating cost, $c_{rel,c+o}$ (*Equation 5*). We define the relative growth rate as:

$$Relative\ growth\ rate = \frac{\tau_D}{\tau_s} \tag{11}$$

where $\tau_D$ is the cell division time in the absence of flagella and the supply of nutrients is by diffusion only, and $\tau_s$ is the cell division time in the presence of a flagellum and nutrient supply is increased by swimming. Assuming that cell division time is limited by nutrient supply, the cell division times can be calculated from the cost of the cell and the nutrient uptake rate.

$$\tau_D = \frac{c_D}{n_D}, \tau_s = \frac{c_s}{n_s} \tag{12}$$

Here, $c_s$ and $c_D$ are the cell costs with and without the flagella, and $n_s$ and $n_D$ are the nutrient uptake rates with and without flagella. *Equation 12* can be substituted into *Equation 11* to obtain:

$$Relative\ growth\ rate = \frac{c_D/n_D}{c_s/n_s} = \frac{c_D n_s}{c_s n_D}, \tag{13}$$

and since the Sherwood number is equal to the ratio of the nutrient uptake rates we find:

$$Relative\ growth\ rate = \frac{c_D}{c_s} Sh. \tag{14}$$

Next, the cell costs are normalised so $c_D = 1$ (just the cell) and $c_s = 1 + c_{rel,c+o}$ (cell + flagella) yielding *Equation 5*.

## Derivation of scaling relations from Stokes' law

Here, we derive, under various assumptions, two power law relations from Stokes' law: (1) a relation between cell volume and the relative flagellar construction cost in eukaryotic flagellates, and (2) a relation between cell volume and swimming speed. Stokes' law describes the power, $P$, that is needed to propel a sphere of radius, $r$, through a liquid of viscosity, $\eta$, at a speed, $v$:

$$P = 6\pi\eta r v^2 \tag{15}$$

Since we are dealing with spheres, we can also express *Equation 15* in terms of volume, $V$, as:

$$P = 6\pi\eta \left(\frac{3}{4\pi}\right)^{\frac{1}{3}} V^{\frac{1}{3}} v^2. \tag{16}$$

The first power law we derive is for the relation between cell volume and relative flagellar construction cost that is observed for the eukaryotic flagellates in *Figure 1B*. We do so for the special case in which the swimming speed doesn't change with cell volume. This allows us to simplify *Equation 16* to:

$$P = a V^{\frac{1}{3}}. \tag{17}$$

Here, $a$ is used to indicate that there is a proportionality constant. It has no particular value and can't be compared between, or within equations. We assume that the absolute construction cost, $c_{abs}$, of a flagellum is linearly related to its power required for swimming (see below), so we have:

$$P = ac_{abs} \tag{18}$$

The absolute flagellar construction cost is related to the relative flagellar construction cost by **Lynch and Marinov, 2015**:

$$c_{abs} = c_{rel}aV^{0.97\pm0.04}. \tag{19}$$

Now, we combine **Equations 17–19** to obtain the relation between cell volume and relative flagellar construction cost (**Equation 6**).

The reasoning behind flagellar operating cost being linearly proportional to flagellar construction cost is as follows. In the case of the eukaryotic flagellum the flagellum length is linearly proportional to the number of motors. So, a doubling of flagellar length (which doubles the construction cost), either by having two flagella or by doubling the length of the existing flagellum, also doubles the energy consumption. In the case of bacteria, the same argument can be made if doubling of construction cost means having two flagella, which would also double energy consumption. The effect of doubling the length of a bacterial flagellum is not so obvious. However, doubling the length of the bacterial flagellum presumably doubles the amount of water that is displaced per rotation, requiring a doubling of the operating cost. Interactions between flagella or between parts of the same flagellum, either directly or hydrodynamically, probably affect the operating cost. So, the linearity between operating and construction cost should be viewed as an approximation. There is also limited empirical support for the linear proportionality of flagellar operating and construction cost. The operating cost per μm flagellum in *Chlamydomonas* and *Paramecium* is similar (Results), despite them having a wildly different number of flagella.

The second power law we derive is for the relation between the cell volume and the swimming speed that is observed in **Figure 2A**. We start with **Equation 16**, and substitute the power, $P$, by the flagellar operating cost, $c_{oper}$:

$$c_{oper} = 6\pi\eta \left(\frac{3}{4\pi}\right)^{\frac{1}{3}} V^{\frac{1}{3}} v^2. \tag{20}$$

The flagellar operating cost is assumed to be a fixed fraction of the total cell operating (or maintenance) cost (see below), so that we can substitute the operating cost by a constant times the total cell operating cost, $c_{oper,cell}$:

$$ac_{oper,cell} = 6\pi\eta \left(\frac{3}{4\pi}\right)^{\frac{1}{3}} V^{\frac{1}{3}} v^2. \tag{21}$$

The total cell operating cost is related to the cell volume (**Lynch and Marinov, 2015**), so we have:

$$c_{oper} = ac_{oper,cell} = aV^{0.88\pm0.07}. \tag{22}$$

Combining **Equations 21 and 22** yields:

$$aV^{0.88\pm0.07} = 6\pi\eta \left(\frac{3}{4\pi}\right)^{\frac{1}{3}} V^{\frac{1}{3}} v^2. \tag{23}$$

Simplifying this yields the relation between cell volume and swimming speed (**Equation 7**).

We want to compare **Equation 7** to the data in **Figure 2A**. However, **Equation 7** only holds when the flagellar operating cost, $c$ , is a fixed fraction of the total cell operating (or maintenance) cost. This is true if both the flagellar operating cost and the total cell operating cost scale with cell volume in the same way. Which is what we will demonstrate here. However, we don't have the flagellar operating cost data over a range of cell volumes. So, instead, we start by assuming that the flagellar *operating* cost is linearly proportional to the flagellar *construction* cost (justified above). Next, we determine the power law relation between cell volume and absolute flagellar construction cost for the subset of

species for which we also have swimming speed data (all species in *Figure 2A*, but only a subset of the species in *Figure 1A*). This power law has an exponent of 0.86±0.05. We compare this to the empirically determined exponent in the power law relation between cell volume and total cell operating cost (*Lynch and Marinov, 2015*), 0.88±0.07. Because the exponents are the same, we can conclude that the flagellar construction cost, and thereby the flagellar operating cost, is indeed a fixed fraction of the total cell operating cost. Thus, it is safe to compare *Equation 7* to the data in *Figure 2A*.

## Acknowledgements

We thank Sergio A Muñoz-Gómez and Bogoljub Trickovic for critically reading and commenting on the manuscript. This work was supported by the US Department of Army, MURI award W911NF-14-1-0411; the National Institutes of Health, R35-GM122566-01; the National Science Foundation, DEB-1927159; and the Moore-Simons Project on the Origin of the Eukaryotic Cell, Simons Foundation 735927, https://doi.org/10.46714/735927.

## Additional information

### Funding

| Funder | Grant reference number | Author |
| --- | --- | --- |
| Simons Foundation | 735927 | Paul E Schavemaker Michael Lynch |
| US Department of Army | W911NF-14-1-0411 | Michael Lynch |
| National Institutes of Health | R35-GM122566-01 | Michael Lynch |
| National Science Foundation | DEB-1927159 | Michael Lynch |

The funders had no role in study design, data collection and interpretation, or the decision to submit the work for publication.

### Author contributions

Paul E Schavemaker, Conceptualization, Data curation, Formal analysis, Investigation, Methodology, Validation, Visualization, Writing - original draft, Writing - review and editing; Michael Lynch, Conceptualization, Funding acquisition, Investigation, Project administration, Supervision, Writing - review and editing

### Author ORCIDs

Paul E Schavemaker  http://orcid.org/0000-0002-8579-8802

### Decision letter and Author response

Decision letter https://doi.org/10.7554/eLife.77266.sa1
Author response https://doi.org/10.7554/eLife.77266.sa2

## Additional files

### Supplementary files

• Transparent reporting form

### Data availability

The collected data (with references) are reported in source data files.

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
