## [Editor Report]

This work demonstrates convincingly that energetic considerations (building costs versus potential benefit) must be taken into account to understand the evolution of flagella. It provides compelling evidence to the long-standing question of why bacteria, archaea, and eukaryotes evolved with different types of flagella.

---

## [Decision Letter]

**Decision letter after peer review:**

Thank you for submitting your article "Flagellar energy costs across the Tree of Life" for consideration by *eLife*. Your article has been reviewed by 2 peer reviewers, and the evaluation has been overseen by a Reviewing Editor and Aleksandra Walczak as the Senior Editor. The following individuals involved in review of your submission have agreed to reveal their identity: Etienne Loiseau (Reviewer #1).

Essential revisions:

Both reviewers are quite enthusiastic about your study. Beyond some comments and requested text changes, you will find that Reviewer #2 questions several missing structural elements in the calculation of the construction cost of eukaryotic flagella. As these considerations may affect your estimates and conclusions, we would like you to take these comments into consideration to see if your estimates need to be revised or not.

*Reviewer #1 (Recommendations for the authors):*

1) In the last paragraph of the "Bacteria – cost of the *E. coli* flagellum" section. While some hypotheses are discussed to explain the apparent contradiction between flagellum operating cost and cell operating cost, it would be interesting to discuss whether this apparent discrepancy could modify the conclusion and how.

2) Discussion section, "speed increases weakly with cell volume…except perhaps for eukaryotic flagellates…" I agree that the swimming speed plotted in Figure 2a seems to be independent of the cell volume for eukaryotic flagellates, but this is also the case for bacteria. Data are fitted with an empirical power law, but it seems that each group (bacteria, eukaryotic flagellates and ciliated cell) have a characteristic range of swimming speed.

3) Figure 2C and 2D, I think that Figure 2D is much more informative than Figure 2C. Indeed, the power fit for bacteria in Figure 2c seems a bit weak as the data are not spanning more than one decade in cell volume. On the contrary, the representation of the swimming speed per ATP in Figure 2D is robust with all the data falling on a master curve.

4) On page 13, "cells with larger volumes…" line 24-27. I'm a bit sceptical with the argument that the larger number of flagella on larger cells could induce interferences between flagella and in turn reduce efficiency. Instead, there are many theoretical/numerical studies (for example works from the groups of Golstein, Golestanian, Gompper) that show that hydrodynamic interaction in dense carpet of cilia/flagella are responsible for the emergence of metachronal waves that are energetically more efficient to transport fluid.

*Reviewer #2 (Recommendations for the authors):*

My main concern about this work is related to the estimated construction cost of eukaryotic flagella as described in the text. A few points that the authors should take into account in their calculations:

1. While structural studies have identified several proteins and protein complexes that bind along the axoneme in a periodic manner, mass-spec studies have shown that over 600 different proteins form the eukaryotic flagella (Pazour and Witman, JCB 2005). From the authors' description, it is hard to understand what database they have used for their analysis (A full list of identified flagellar proteins in *Chlamydomonas* can be found at http://chlamyfp.org/). Additionally, until recently the exact structure of the central apparatus was unknown. The structure was recently resolved by the Zhang lab (https://doi.org/10.1101/2022.01.23.477438).

2. In their construction cost the authors include 279 IFT complexes per flagellum. From a quick look in Vannuccini et al., 2016 it is not clear how the authors got this number. Is it the total number of trains that are used to build the flagellum? Is it just long trains after 120 min? if so, what about the short trains that participate in the regeneration process? Furthermore, the IFT system is required for the maintenance of the flagella. In the conditional fla10-1 mutant, the incorporation of tubulin at the distal end of the flagella is decreased after a shutdown of the IFT system and the flagella resorb (Marshall and Rosenbaum, JCB 2001; Vannuccini et al., JCS 2016), which means that construction happens also after 120 min.

3. The ciliary microtubules are heavily decorated with various post-translational modifications (Orbach and Howard, Nat Commun 2019). The authors ignore this part in their construction cost calculation. The authors should elaborate what is the contribution of PTMs to the construction cost of the flagella.

4. *Chlamydomonas*, as well as other organisms, have a cap at the distal tip that connects the axoneme structure to the membrane (Dentler, JCS 1980). What is the construction cost of the structure?

---

## [Author Response]

Reviewer #1 (Recommendations for the authors):1) In the last paragraph of the "Bacteria – cost of the *E. coli* flagellum" section. While some hypotheses are discussed to explain the apparent contradiction between flagellum operating cost and cell operating cost, it would be interesting to discuss whether this apparent discrepancy could modify the conclusion and how.

It is not clear which conclusion the reviewer is referring to. Of all the conclusions listed in our conclusion section, the operating cost of the bacterial flagellum bears directly on only one, the benefit of swimming in a homogenous medium (Figure 2F).

The apparent mismatch of flagellum and cell operating cost could be caused by overestimating the flagellar cost or underestimating the cell operating cost. If the flagellar operating cost is overestimated, the conclusion that small cells (mostly bacterial cells in our dataset) don’t benefit from a flagellum in a homogenous medium remains the same. Even if the flagellar operating cost is removed completely from the cost in equation 5, leaving only the flagellar construction cost, the relative growth rate increases but is still below 1 for most bacterial species. (The relative growth rate is determined relative to the cell without the flagellum. If the flagellum-carrying cell grows faster that the cell without a flagellum, the relative growth rate is >1.) An underestimation of cell operating cost means that our estimate of the relative cost of flagella is too high. However, for this to have any significant effect on the results presented in Figure 2F the cell operating cost would have to be underestimated by one or two orders of magnitude, which seems unlikely. The reason being that the contribution of the operating cost to the whole cell budget is small (see Results section).

A statement about this has now been added to the discussion.

2) Discussion section, "speed increases weakly with cell volume…except perhaps for eukaryotic flagellates…" I agree that the swimming speed plotted in Figure 2a seems to be independent of the cell volume for eukaryotic flagellates, but this is also the case for bacteria. Data are fitted with an empirical power law, but it seems that each group (bacteria, eukaryotic flagellates and ciliated cell) have a characteristic range of swimming speed.

The reason that eukaryotic flagellates have their own regression and the others do not is because we made a prediction about the swimming speed of eukaryotic flagellates (in the section “Scaling properties…”). The way we phrased it, it may have seemed that we were saying that for both bacteria and ciliates, taken separately, the swimming speed increases with cell volume. This wasn’t our intended meaning and we have rephrased the sentence.

3) Figure 2C and 2D, I think that Figure 2D is much more informative than Figure 2C. Indeed, the power fit for bacteria in Figure 2c seems a bit weak as the data are not spanning more than one decade in cell volume. On the contrary, the representation of the swimming speed per ATP in Figure 2D is robust with all the data falling on a master curve.

We have removed figure 2C.

4) On page 13, "cells with larger volumes…" line 24-27. I'm a bit sceptical with the argument that the larger number of flagella on larger cells could induce interferences between flagella and in turn reduce efficiency. Instead, there are many theoretical/numerical studies (for example works from the groups of Golstein, Golestanian, Gompper) that show that hydrodynamic interaction in dense carpet of cilia/flagella are responsible for the emergence of metachronal waves that are energetically more efficient to transport fluid.

This has been amended.

Reviewer #2 (Recommendations for the authors):My main concern about this work is related to the estimated construction cost of eukaryotic flagella as described in the text. A few points that the authors should take into account in their calculations:1. While structural studies have identified several proteins and protein complexes that bind along the axoneme in a periodic manner, mass-spec studies have shown that over 600 different proteins form the eukaryotic flagella (Pazour and Witman, JCB 2005). From the authors' description, it is hard to understand what database they have used for their analysis (A full list of identified flagellar proteins in Chlamydomonas can be found at http://chlamyfp.org/). Additionally, until recently the exact structure of the central apparatus was unknown. The structure was recently resolved by the Zhang lab (https://doi.org/10.1101/2022.01.23.477438).

We have examined the paper by (Pazour and Witman, JCB 2005) as well as a more recent cilia proteomics paper (Yano et al., J. Proteomics, 2012). Unfortunately for us, both studies identify flagellar proteins, but don’t quantify them. There are quantitative proteomics studies available for whole *Chlamydomonas* cells, e.g. from PaxDB and (Muller et al., Nature 2020). However, we can’t reliably use this information even if we knew what proteins were identified to be in, or part of, the flagellum. First, not all proteins identified in the flagellum necessarily contribute to the flagellar functions, as there presumably is leakage from the cytoplasm into the flagellum. Second, even if the identified protein contributes to flagellar function, it may also play a role in other parts of the cell (for example tubulin). Third, for all of the proteins that are in the axonemal structure, there may be a pool of subunits necessary for keeping the flagellum stable (because of mass action), but these don’t contribute to the flagellar mechanism per se. Whether one should count these is unclear and presumably depends on what question you want to answer.

To avoid the complications just mentioned we opted for determining the number of proteins based on structural information. We were able to obtain structural data for every major contributor to the mass of the axoneme. The sources for all proteins/complexes that are included in our costs are mentioned in the main text as well as in “Table 1-source data 1”, and we have no reason to think that any major structural elements are missing. For some of these complexes, the exact protein composition wasn’t known, so we used the volumes of these complexes that were determined by lower resolution EM data. This is also what we used to determine the cost of the protein complexes that are associated with the central pair of MTs. The volumes were converted into the number of (average) amino acids using the conversion factor explained in the Methods. We have not included any soluble proteins that are present in the flagellum as these aren’t visible in the structures. The possible impact of those missing proteins on the cost of the flagellum is discussed next.

Another argument, based on volume occupancy of proteins in the flagellum space, suggests that our eukaryotic flagellum cost is close to the real one. Adding up the volumes of all proteins in the axoneme, the IFT particles, and the membrane protein domains that we assume point inwards, we come to a total protein volume fraction of 18%. For the flagellum internal volume we used length = 11 μm and width = 0.24 μm (0.25 – 2x0.005 for the membranes). This volume fraction is higher than the macromolecular volume fraction of the *E. coli* cytoplasm, 0.16 (Konopka et al., J. Bacteriol., 2009). If the remaining free volume (0.82) were to be occupied by a fraction of 0.16 of proteins, then we would have to add an additional 46% to the cost of the flagellum. However, it is unlikely that this amount of extra protein can be packed into the remaining volume without severely impacting the mobility of proteins in the whole flagellum.

We have added some of these considerations to the methods section.

2. In their construction cost the authors include 279 IFT complexes per flagellum. From a quick look in Vannuccini et al., 2016 it is not clear how the authors got this number. Is it the total number of trains that are used to build the flagellum? Is it just long trains after 120 min? if so, what about the short trains that participate in the regeneration process? Furthermore, the IFT system is required for the maintenance of the flagella. In the conditional fla10-1 mutant, the incorporation of tubulin at the distal end of the flagella is decreased after a shutdown of the IFT system and the flagella resorb (Marshall and Rosenbaum, JCB 2001; Vannuccini et al., JCS 2016), which means that construction happens also after 120 min.

The number of IFT complexes is an average of the sum of long and short IFT particles over all the time points in Figure 1E. These numbers refer to the number of IFT particles being present along the whole flagellum at a particular timepoint. Note also that as far as the total flagellar cost is concerned, the construction cost of the IFT particles adds only 1%, so any inaccuracies are not very relevant to the conclusions we draw in the paper.

3. The ciliary microtubules are heavily decorated with various post-translational modifications (Orbach and Howard, Nat Commun 2019). The authors ignore this part in their construction cost calculation. The authors should elaborate what is the contribution of PTMs to the construction cost of the flagella.

We have looked at posttranslational modifications for the archaeal flagellar costs. There are a considerable number of sugar groups (35 per archaellin) that are added to the archaellins that make up the flagellar filament. There this modification adds ~17% to the flagellum costs (assuming all flagellins have a full complement of these sugars). The exact amount of biomass added to the tubulin by posttranslational modification is unclear, it is not mentioned in (Orbach and Howard, Nat Commun 2019). There are several considerations that suggest that the posttranslational modification of tubulin doesn’t contribute as much (relative) biomass to the eukaryotic flagellum as is the case for the archaeal flagellum. First, the number of glutamines added in a polyglutamylation appears to vary between 1-20 per tubulin (Janke et al., EMBO reports 2008), which constitutes a lower cost than the 35 sugar units added to archaellin. The other modifications appear to add even less biomass. Second, the tubulin monomer is larger (~450 AA) than archaellin (207 AA), reducing the relative cost of the modification. Third, not all tubulin monomers are modified (the A tubule contains mostly unmodified tubulin) (Wloga et al., Cold Spring Harb Perspect Biol 2017). Fourth, the contribution of tubulin to total eukaryotic flagellum cost (~30%) is smaller than that of archaellin to the archaeal flagellum cost (~98%). Together these considerations suggest a cost of posttranslational modifications that is ~50-fold lower than the posttranslational modifications in the archaeal flagellum, contributing < 1% to eukaryotic flagellum costs.

We have added some of these considerations to the methods section.

4. Chlamydomonas, as well as other organisms, have a cap at the distal tip that connects the axoneme structure to the membrane (Dentler, JCS 1980). What is the construction cost of the structure?

From Figure 20 in the Dentler paper it appears that the cap is about as long as the flagellum is wide (0.25 um). This would thus add just a fraction of 0.25 um/11 μm = 1/44 to the length of the flagellum, which is negligible. However, note that because in our model of the eukaryotic flagellum, the axoneme runs all the way to the tip, taking the place of the cap, we have already taken the cap into account implicitly.